# Polymer Thermal Treatment Production of Cerium Doped Willemite Nanoparticles: An Analysis of Structure, Energy Band Gap and Luminescence Properties

**DOI:** 10.3390/ma14051118

**Published:** 2021-02-27

**Authors:** Ibrahim Mustapha Alibe, Khamirul Amin Matori, Mohd Hafiz Mohd Zaid, Salisu Nasir, Ali Mustapha Alibe, Mohammad Zulhasif Ahmad Khiri

**Affiliations:** 1Material Synthesis and Characterization Laboratory (MSCL), Institute of Advanced Technology (ITMA), Universiti Putra Malaysia (UPM), Serdang 43400, Selangor, Malaysia; 2National Research Institute for Chemical Technology Zaria, Zaria P.M.B. 1052, Kaduna State, Nigeria; 3Department of Physics, Faculty of Science, Universiti Putra Malaysia (UPM), Serdang 43400, Selangor, Malaysia; mzulhasif@gmail.com; 4Department of Chemistry, Faculty of Science, Federal University Dutse, Dutse P.M.B. 7156, Jigawa State, Nigeria; salisunasirbbr@gmail.com; 5Mechanical Engineering Departments, Federal Polytechnic Damaturu, Damaturu 620221, Yobe State, Nigeria; alibebenisheikh77@gmail.com

**Keywords:** willemite, cerium, calcination, nanoparticles, doping, optical properties, band gap energy

## Abstract

The contemporary market needs for enhanced solid–state lighting devices has led to an increased demand for the production of willemite based phosphors using low-cost techniques. In this study, Ce^3+^ doped willemite nanoparticles were fabricated using polymer thermal treatment method. The special effects of the calcination temperatures and the dopant concentration on the structural and optical properties of the material were thoroughly studied. The XRD analysis of the samples treated at 900 °C revealed the development and or materialization of the willemite phase. The increase in the dopant concentration causes an expansion of the lattice owing to the replacement of larger Ce^3+^ ions for smaller Zn^2+^ ions. Based on the FESEM and TEM micrographs, the nanoparticles size increases with the increase in the cerium ions. The mean particles sizes were estimated to be 23.61 nm at 1 mol% to 34.02 nm at 5 mol% of the cerium dopant. The optical band gap energy of the doped samples formed at 900 °C decreased precisely by 0.21 eV (i.e., 5.21 to 5.00 eV). The PL analysis of the doped samples exhibits a strong emission at 400 nm which is ascribed to the transition of an electron from localized Ce_2f_ state to the valence band of O_2p_. The energy level of the Ce^3+^ ions affects the willemite crystal lattice, thus causing a decrease in the intensity of the green emission at 530 nm and the blue emission at 485 nm. The wide optical band gap energy of the willemite produced is expected to pave the way for exciting innovations in solid–state lighting applications.

## 1. Introduction

The contemporary advancement in organic and inorganic semiconductor materials for optoelectronic uses especially in optical data storage, plasma display panels, optical amplification, sensing, and laser devices have garnered considerable interests in the past 20 years or so from scientists and researchers in material science and engineering. Over the decades, the advancement of conjugated organic semiconductor nanomaterials has led to potential applications in optoelectronic devices, such as organic LEDs. Conjugated organic materials such as polymers in their pure state are usually semiconductors or insulators, possessing lower conductivity. The band-gap energy of organic conjugated semiconductors is related to the electronic band structure and depends on the synthesis conditions such as temperature [1]. They tend to decrease with increasing temperature due to increased atomic vibrations that lead to larger average interatomic spacing. When these materials are doped, charge carriers are created and move along the polymer chains. As a result, the conductivity remarkably increases by several orders of magnitude [2]. However, their main drawbacks in the field of optoelectronics when compared to their inorganic counterpart are low optimum temperature operation windows. In addition, they have a tendency to be readily affected by ionizing radiation and the possibility to have a lower life span due to the more complex chemistry involved [1,2].

An inorganic semiconductor is mainly compounds made from noncarbon-based materials. These materials can be categorized roughly into nitrides, phosphate, fluoride, aluminates, and silicates. Nitride materials such as titanium and zirconium nitride have been studied to serve as an alternate plasmonic material in the near-IR and visible ranges because of their low intrinsic loss, semiconductor-based design, compatibility with standard nanofabrication processes, and tenability [1]. Inorganic phosphate based materials, like phosphate glasses, comprises numerous optical, physical, and chemical properties such as wide energy band gap, higher absorption in the UV region, lower lowest transition temperature and desirable rheological characteristics and excellent chemical stability [3,4]. Fluoride-based phosphors offer advantages such as higher ability to dissolve larger amounts of dopants and lesser phonon energy. These materials are currently studied as potential luminescent materials owing to their capabilities for multitudinous applications in photonic and biophotonic domains [5,6]. Aluminate-based materials such as aluminum titanate are considered as one of the novel photocatalytic materials because of their ability to accelerate photoreaction [7]. Among the most widely studied inorganic semiconductors for optoelectronics application, silicate based materials are well-thought-out for their outstanding features or characteristics, for example, good physical and chemical stability, great thermal stability, water resistance ability, and excellent mechanical properties [8,9,10]. In this light, a specific interest has been developed for the production of zinc silicates [11,12,13,14]. Zinc silicate or willemite (Zn_2_SiO_4_) as often called is one of the famous silicate-based luminescent materials that was discovered over two centuries ago occurring freely in nature in association and or with minor ores of zinc minerals. Structurally, the compound possesses a phenacite configuration or form in which Zn–O tetrahedral and Si–O tetrahedral shares angles, creating hollow tubes that act parallel to [0001] plane [15,16]. Willemite has several prospective applications such as photonic devices [17,18], adsorbents [19,20], and laser crystal systems [21,22]. Willemite is also serving as a host material for guest ions like transitional metals or rare earth metals due to its ability to display a wide range of multicolors for luminescence devices [11,23,24,25,26,27,28]. For instance, the luminescence properties of rare earth ions (RE^3+^) such as the longer emission lifetime, sharper luminescence array, and the characteristic spectral features (owed to their filled 4f shells and spared by the exterior 5s^2^ and 5p^6^ orbitals) are some of the exciting factors for choosing them in modern day optoelectronic applications. In this light, willemite doped with RE^3+^ is attracting significant attention for its luminescence efficiency and color purity for lighting devices [29,30].

Willemite doped with RE^3+^ has been synthesized by several developed techniques such as the conventional sol–gel and solid state method [26,27,28,31,32]. Other methods include hydrothermal [33], pyrolysis method [34], sol–emulsion–gel method [35], microwave heating [36], and combustion method [37]. It is interesting to note that sol–gel method is one of the prominent techniques for willemite dope with RE^3+^ phosphor materials, although the temperature involved in the heat-treatment and the inability to control particles agglomeration is still a matter of concern to must researchers as these affect the optical properties of the materials. In this sense, Babu et al. reported the synthesis of Ce^3+^: Zn_2_SiO_4_ phosphors material at an optimum sintering temperature of 1000 °C via sol–gel technique [38]. Similarly, Chelouche et al. produced SiO_2_/Zn_2_SiO_4_:Ce nanophosphors under higher processing or calcination temperature of 1200 °C [39]. On the other hand, the synthesis of willemite doped with RE^3+^ by solid state method often involves greater annealing temperature than the conventional sol–gel method [27]. This is attributed to the tedious procedures such as melt–quenching practice at a higher temperature of 1400 °C [26,27,32]. Willemite based phosphor produced by Omar et al. [28], employs melt–quenching of the precursor at 1400 °C and sintering between 800 and 1000 °C.

However, most of these methods have limitations such as complicated procedures, the lengthier aging period for several hours, the toxic reagent used as surfactants or catalyst, and higher heat-treatment conditions which cause greater energy demand. It is important to note that phosphor industries alongside other manufacturing companies have been criticized for high energy consumption and carbon emission to the atmosphere [40]. These challenges could be mitigated by applying a polymer thermal treatment technique. This method offers dynamic means in the synthetic chemical approach to fabricate a novel nano-sized metal oxide inorganic semiconductor inexpensively nowadays [41,42,43,44]. Although, the synthesis of undoped willemite by polymer thermal treatment method has been reported elsewhere by the authors [45], nevertheless, to the best of our knowledge, no literature discoveries or reports are yet available on the production of willemite doped with RE^3+^ utilizing this technique. Hence, this research work is focused on the fabrication of willemite Ce^3+^ nanoparticles (NPs). Moreover, the influence of the calcination temperature and the effects of Ce^3+^ doping content on the microstructural and optical characteristics of the willemite NPs have also been thoroughly investigated and reported here.

## 2. Materials and Methods

The polymer (polyvinyl pyrrolidone PVP 2900 Mw) was used as a capping agent to stabilize the NPs and reduce agglomeration. It was procured from Sigma Aldrich (Darmstadt, Germany). Metallic precursors of zinc acetate dihydrate, [Zn(CH_3_COO)_2_.2H_2_O] Mw = 219.49 g/mol), silicon tetraacetate, [Si(OCOCH_3_)_4_] (Mw = 264.26 g/mol), and cerium(III) nitrate hexahydrate reagent [Ce(NO_3_)_3_.6H_2_O] (Mw 434.22 g/mol) with their purity exceeding 99% were also purchased from Sigma Aldrich. Deionized water with resistivity and conductivity of 18.2 MΩcm^–1^ and 0.055 μScm^–1^ was used as a solvent to dissolve the precursors. No further chemical purification was conducted. Thus, all chemicals were used as obtained.

In brief, undoped willemite was prepared by forming an aqueous solution of PVP. The solution was made by dissolving 4 g of PVP in 100 mL of deionizing water in a beaker using a magnetic stirrer. An equal amount of the metallic precursor Zn(CH_3_COO)_2_.2H_2_O and (Si(OCOCH_3_)_4_ (0.1 mmol each) was added to the PVP solution and stirred for 2 h. The solution was then poured into a clean glass petri dish, and the water was evaporated at 80 °C in an electric oven overnight. The resultant yellowish material acquired was pulverized into fine particles using pestle and mortar. The powder then underwent a calcination process at temperatures range between 500 and 1000 °C for 3 h for crystallization of willemite NPs and elimination of the organic components in the samples [45]. The same procedure was followed for the preparation of willemite: Ce^3+^ with a molar ratio of Ce_X_:Zn_1−X_:Si_1_ (x = 0, 1, 2, 3, 4, and 5 mol%). The schematic diagram for the synthesis process is presented in Figure 1.

## 3. Characterization

In this work, various analytical techniques were used for this study. The thermal analysis of the material was conducted under nitrogen conditions to study the decomposition behavior of metallic precursor combined with the polymer (PVP). This was achieved using TGA/DSC 1HT model, (Mettler Toledo, Shah Alam, Selangor, Malaysia). The structural features of the samples such as the degree of crystallinity and willemite phase formation were examined using an X-ray diffraction spectrometer (XRD Shimadzu model 6000 Lelyweg1, Almelo, The Netherlands). A radiation source of Cu kα (0.154 nm) was used in generating the diffraction peaks within an angle range of 2θ equal to 10–80°. The willemite crystalline size was evaluated for the most intense peak using the famous Scherer’s equation expressed mathematically in Equation (1) below:(1)                 D=0.9λβcosθ
where *D* is defined as the crystallite size of the material given in nanometer (nm), the Bragg’s angle and the X-ray wavelength are denoted by *θ* and *λ*, respectively. While *β* denotes the diffraction full width at half maximum intensity. The functional groups and the kind of bonding formation that exists in samples were investigated using infrared spectroscopy (FT–IR, Perkin Elmer model 1650, Labexchange, Swabian Burladingen, Germany). Raman spectroscopy was employed to examine the chemical composition and structures of the samples. The analysis was conducted before and after the calcination process using a ‘Wissenschaftliche Instrumente und Technologie’ (WITec) Raman spectrometer, Alpha 300R (WITec GmbH, Ulm, Germany). The Raman frequency was developed within an integration period of 5.036 (s) and an excitation wavelength of 532 nm. The NPs size distribution, crystal lattice measurement, and the selected area electron diffraction were viewed using the high-resolution transmission electron microscopy (JEOL HR–TEM model 3010, Tokyo, Japan) with an accelerating voltage of 200 kV. The morphology of the samples was viewed using field electron scanning microscopy (FESEM) equipped with electron disperse X-ray diffraction (EDX) using an FEI Nova NanoSEM 230 (FEI, Hillsboro, OR, USA). The system was operated at accelerating voltage of 5 kV. The light absorption features of the materials were analyzed using the UV–vis spectrometer (Shimadzu model UV–3600, Kyoto, Japan). The samples were analyzed in powdered form at room temperature. Based on the absorption analysis, the optical band gap energy was estimated by using the Mott and Davis model [45]. The model is stated in the following function:(2)$${\left(\propto hv\right)}^{1/n}=A\left(hv-E_{opt}\right)$$
where A is a constant, the optical energy band gap is denoted by $E_{opt}$, and hv represents the photon energy. A graph of the characteristics ${\left(\propto hv\right)}^{1/n}$ is plotted on the Y-axis as a function of the photon energy on the X-axis. As the function ${\left(\propto hv\right)}^{1/n}$ is approaching value equal to zero, the $E_{opt}\;$is obtained by extrapolation of the X-axis. The value of *n* determines the nature of the transition; whether the transition is direct allowed or forbidden (*n* value is equal to 1/2 or 3/2) or indirect transitions allowed or forbidden, (*n* values is equal to as 2 or 3). The optical emission properties of the samples (in powdered form) were analyzed at room temperature, within the wavelength of 200–800 nm using Photoluminescence (PL) (Perkin Elmer LS 55, Waltham, MA, USA).

## 4. Results and Discussion

### 4.1. Thermogravimetric Analysis

The main purpose of this analysis is to appraise the thermal decomposition properties of the sample before the thermal treatment process and to choose the appropriate calcination temperature for producing the willemite NPs. The thermogravimetric measurement and its derivative (TG–DTG) presented in Figure 2 designates the percentage of weight-loss as a function of the temperature for the sample (metallic salt embedded in PVP) under a nitrogen atmosphere condition at the heating rate of 10 °C/min. The spectra reveal three different decomposition paths. The first weight loss peak observed at 78 °C is recognized to be the moisture content entrapped in the sample due to the hygroscopic nature of PVP [45]. The second point (witnessed a slight weight loss) detected roughly at a temperature of 205 °C is related to the volatile components and owed to the ester group found in the PVP [46,47]. The third stage of the decomposition is the weight loss peak detected at 423 °C which entails decomposition of substantial amounts of the C–C bond in the polymeric chain. The weight loss observed between 450 and 800 °C, which constitutes about 7–8%, may be attributed to the decomposition of remnant of the polymeric chain in the PVP [41,42]. The curve discloses no additional mass loss as the temperature attends 800 °C; at this stage, an entire decomposition of PVP and crystallization from the sample is achieved [41,42,43,44].

### 4.2. X-ray Diffraction Analysis

The XRD analysis of the uncalcined sample at ambient temperature (Figure 3) exhibited a broad diffractogram which represents amorphous characteristics as there are no available peaks that suggest crystallinity. This tallied with the author’s findings elsewhere when willemite precursor was embedded with PVP as capping agent [45]. However, the undoped samples were calcined at a temperature between 500 and 1000 °C shown in Figure 4a. As for the samples that were calcined at 500 and 600 °C, respectively, the diffractogram exhibited sharper diffraction peaks which clearly suggest the crystallinity and the wurtzite structural formation of ZnO that has been ascribed to JCPDS No 01–079–2205. The calcination temperature was further raised to 700 °C, where the spectrum reveals phases of ZnO and SiO_2_ with ICSD No. 170533 corresponding to the latter. It is fascinating to note that SiO_2_ shows an amorphous characteristics at calcination temperature less than 700 °C [41]. According to Zaid et al. [23], the crystallization formation of SiO_2_ phase in ZnO–SiO_2_ system could not be achieved at a temperature below 700 °C due to its amorphous state, and this conforms with the findings of Omar et al. [28]. In a similar context, Babu et al. [48] reported that at greater calcination temperatures beyond 700 °C, the ZnO atoms are the main diffusing component in ZnO/SiO_2_ composite. Thus, ZnO atoms being at the surface diffuses towards the silica matrix and induces the formation of Zn_2_SiO_4_. Therefore, when the calcination temperature was raised up to 800 °C, the spectrum revealed the formation of willemite phase. At this point, all the diffraction peaks are indexed to the standard pattern of α–Zn_2_SiO_2_ (JCPDS No 37–1485) having a space group of R–3, with cell constant a = b = 13.9470 Å, c = 9.3124 Å, except a trace of unreacted ZnO, which was detected at 2θ = 36.37° [28,45,47,48]. Subsequently, the calcination temperature was raised to 900 and 1000 °C, respectively, where the unreacted phase of ZnO disappeared by reacting to form a complete willemite phase. Likewise, it is fascinating to note that the crystalline peaks intensity of willemite tend to be shaper as the calcination temperature increases up to 1000 °C. Based on the findings by Zaid et al. [12], the intensification or sharpness in the intensity of the diffraction peaks show complete crystallization in the samples, and this is attributed to the heat treatment process. Higher calcination temperature increases the atomic mobility in the material which results in particle size growth and better crystallinity [49]. As can be seen in Figure 4b, willemite was doped with Ce^3+^ using different dopant concentration (1–5 mol%) calcined at 900 °C; the spectrum reveals α–Zn_2_SiO_2_ phase and a phase of CeO_2_ with cubic fluorite structure corresponding to PDF card No: 34–0394 [50]. The undoped sample shows sharper peak intensity as compared to the doped ones. The increase of dopant concentration in the samples renders all the peaks intensity to schematically reduced. This is caused by the expansion of the lattice as a result of the exchange or replacement of larger Ce^3+^ ions for smaller Zn^2+^ ions. Similarly, it is interesting to note that the ionic radii of cerium (1.15 Å) are a bit higher than Zn^2+^ (0.74 Å). According to Omar et al. [28], in a similar phenomenon, the substitution of the dopant ions into the structure of willemite causes lattice mismatching, lattice distortion, and strain of the crystal. Thus, the crystal structure of the willemite will be distorted. The average crystallite size of the undoped and the dope willemite at various concentrations was determined by applying the famous Scherrer’s formula shown in equation 1 while taking into consideration the full width at half maximum (FWHM) of the most intense peaks [51]. The crystallite size of the materials ranges around 18.23–27.40 nm and decreases with the increase in doping concentration. The values obtained corroborated well with the particle size values presented in Table 1.

### 4.3. Fourier Transform Infrared Spectroscopy Analysis

The functional groups and the phase composition of the doped and undoped willemite were examined using FT–IR spectroscopy. To study the chemical interaction of metallic precursor and PVP, the uncalcined sample was analyzed at ambient condition (30 °C) displayed in Figure 5a. The spectrum reveals several absorption peaks at 3414, 2945, 1648, 1428, 1278, 850, and 639 cm^–1^ all being ascribed to the existence of organic materials from the PVP [41,42,44]. Informative details on the FT–IR vibrational bands and the corresponding assignments are provided in a tabular form shown in Table 2. Subsequently, the undoped samples were subjected to a calcination process at a temperature between 500 and 1000 °C, this had enabled the decomposition of the entire PVP and crystallization of the samples as seen in Figure 4a. The spectra for the samples calcined between 500 and 700 °C indicates two strong peaks at wavenumber between 407 and 486 cm^–1^ ascribed to Zn–O symmetric stretching vibration and another peak sighted at wavenumber around 810–876 cm^–1^ which was assigned to Si–O symmetric stretching vibration, respectively. While increasing the temperature, from 800 to 1000 °C, three absorption peaks in each spectrum were noticed. This is attributed to the change in phase of material as recorded in the XRD result [28,47]. In this light, the wave number at 380–375 cm^–1^ is related to Zn–O symmetric stretching vibration [47]. Nonetheless, the peaks that appeared between 578 and 574 cm^–1^ are for Zn–O asymmetric stretching vibration [47]. The vibrational bands observed at 885–889 cm^–1^ are assigned to Si–O [11,12,48]. The slight shift to the lower wavelength in the vibrational bands observed was associated with the increase in the calcination temperature [47].

To study the influence of dopants on the willemite NPs, the samples were prepared at a calcination temperature of 900 °C with various dopant concentrations (1–5 mol% Ce^3+^), presented in Figure 5b. The spectrum reveals that the undoped sample has greater absorption of Si–O vibrational bands, and this owed to the strong chemical affinity of silica [28,32,52,53,54]. As the dopant (1–5 mol% Ce^3+^) is substituted into the host matrix, the stretching mode of Ce–O was expected to appear around 500–550 cm^–1^ as reported by Syed Khadar et al. [55]. However, there were no such peaks on the spectra, and this is correlated to the low concentration of the dopant [28,50]. Nonetheless, the broad absorption peak of Si–O vibrational band seems to decrease due to the addition of the cerium ions into the willemite lattice. According to Omar et al., the addition of rare earth ions zinc silicate system affects the SiO_4_ functional group [28].

### 4.4. Raman Spectroscopy Analysis

Raman spectroscopy analysis has been considered as one of the promising analytical instruments for its vital technical applications in the determination of materials’ molecular structures as phonon phenomena. The Raman effects are observed once there is an interaction occurring between an electron cloud of a sample and external electric field of the monochromic light, thus generating a dipole moment within the molecules depending on its polarizability. Here, the analysis was used to investigate the molecular structural behavior of the prepared material. Before the calcination process, the interaction of the metallic precursor and PVP were analyzed at ambient condition shown in Figure 6. The sample exhibits several vibrational bands linked to organic sources from PVP. The Raman vibrational bands and their corresponding assignments are presented in Table 3. The vibrational band at 758 cm^–1^ and the one at 934 cm^–1^ are assigned to the C–C ring vibration and C–C ring breathing, respectively [56]. The band observed at 1023 cm^–1^ can be assigned to C–C backbone and the band at 1370 cm^–1^ is ascribed to the CH deformation [56,57]. The vibrational band observed at the wavenumber 1665 cm^–1^ is due to C=O, and the one observed at 1494 cm^–1^ is attributed to CH_2_ scissors vibration [56,57]. Upon the calcination process from 500–1000 °C for the undoped samples (Figure 7a), the spectra for the samples prepared within range of 500–700 °C reveals a longitudinal optical (LO) vibration associated with the existence of resonance of exciting phonon energy with an electronic interband transition within the ZnO particle [58]. According to previous literature, the LO mode for ZnO–SiO_2_ system can occur at within 500–700 cm^–1^ wavenumber as it is ascribed to the defects such as oxygen vacancy or interstitial zinc in ZnO [59,60]. Heating from 800 °C and beyond up to 1000 °C caused the formation of crystalline willemite. The spectrum (Figure 7a) of the samples calcined between 800 and 1000 °C reveals that the crystalline willemite possesses vibrational peaks positioned at 866, 906, and 947 cm^−1^ that originates from the surface of siloxane group [61,62]. The Raman intensity rises with the corresponding increment in the calcination temperature where the phenomenon is related to the enhancement in the crystalline behavior of the willemite NPs. To study the influence of the cerium dopants (1–5 mol% Ce^3+^) on the willemite NPs calcined at 900 °C, the spectrum (Figure 7b) reveals the expansion in the peak width at 866, 906, and 947 cm^−1^ and the appearance of new peak at 663 cm^−1^ attributed to the lattice expansion due the presence of Ce^+3^ ions defects [63]. The higher amount of Ce^+3^ causes the intensity of the defects at 663 cm^−1^ to heighten due to factors that affect the lattice expansion as surface stress and phonon confinement [63,64].

### 4.5. TEM Analysis

The TEM analysis shown in Figure 8 was used to study the effect of the calcination temperature and doping concentration on the morphology, shape, and distribution of NPs produced. The undoped samples were calcined at temperature between 500 and 1000 °C shown in Figure 8, the obtained micrograph reveals that samples calcined at 500–700 °C, respectively, exhibits spherical morphology and no sign of crystal growth due to noncrystallization of silica content in the ZnO–SiO_2_ system at that temperature [53,54]. Upon calcination at higher temperatures between 800 and 1000 °C, the samples crystallinity is attained and revealed relatively uniform morphology and good nanoparticle distribution. These findings were found to be in good conformity with XRD result in this work and the findings of other researchers where the particle size is increasing with respect to increasing temperature [48,65,66,67,68]. The effect of dopants on the willemite NPs was investigated on the samples prepared at the calcination temperature of 900 °C with various concentrations of the cerium dopant (1–5 mol% Ce^3+)^ which are presented in Figure 9. These micrographs demonstrate the homogenization of Ce^3+^ in the willemite nanostructure [35]. The nanoparticle size increases with respect to increase in the cerium ion by necking between the neighboring particles. This is attributed to the lattice expansion caused by the substitution of the Ce^3+^ ions whose ionic radii is 1.15 Å, for Zn ions having less ionic radii 0.74 Å, thus distorts the willemite structure [39]. The mean NPs particle size for the doped samples was determined using Image J software (version 1.40g, Madison, WI, USA) shown in Figure 10, which were estimated to be 23.61 nm at 1 mol% to 34.02 nm at 5 mol% of the cerium dopant. According to Omar et al. [32] and Rasdi et al. [53], the exchange interaction between ions in higher dopant concentration may result in particle size growth of the material.

### 4.6. FESEM Analysis

The FESEM equipped with EDX provides information on the external morphology, nanoparticle distribution, and elemental composition contained in the samples. The micrographs of the undoped samples calcined between 500 and 1000 °C are shown in Figure 11. There were no considerable changes observed in the NPs size for the samples synthesized between 500, 600, and 700 °C, respectively. The materials revealed a tiny particle-like morphology. Subsequently, when the calcination temperature is increased between 800 and 1000 °C, larger particle sizes were formed with irregular shape and morphology. This may be related to the rise in the surface energy of the samples where the smaller NPs fused into the neighboring particles, thus forming larger ones [44,48,65]. The effect of the cerium ion with different dopant amounts (1–5 mol% Ce^3+^) on the samples fabricated at the calcination temperature of 900 °C is presented in Figure 12. The images reveal that the micrograph is composed of NPs with the nearly polyhedral shape. It was further observed that with the increase in the cerium ion concentration, the sample revealed densely packed particles of different irregular sizes appearing with strong necking between the neighbor particles. It is interesting to observe that the replacement of Ce^3+^ ions for Zn ions in willemite structure might distort the particle sizes and shapes [25,56]. Similarly, the sample prepared with 4 and 5 mol% shows more heightened edge-connection of the particles.

The purity of the willemite doped samples together with the elemental composition was studied using the EDX technique as depicted in Figure 13. EDX is an analytical technique essentially for the elemental determination composition and chemical characterization of a sample. The basic principle of the analysis relies on the interaction of source of X-ray excitation on the sample. Its characterization ability is related to the fundamental principle that every distinct element possesses a unique atomic structure matching a unique peak on the emission spectrum [69]. The emission spectra are shown in Figure 13, illustrating the EDX of both undoped and the doped samples prepared at 900 °C. The appearance of several peaks matching to Zn, Si, O, and the dopant Ce, ratify the purity of the willemite NPs made and further revealed that there was no elemental loss in the synthesis procedure [38].

### 4.7. UV–Vis Analysis

The effects of the calcination temperatures and doping concentration on the light absorption properties of the synthesized willemite NPs were appraised using UV–vis absorption spectroscopy. In this regard, the absorption spectra of the undoped samples treated at different temperatures between 500 and 1000 °C were analyzed as portrayed in Figure 14a. The absorption spectra for the willemite calcined at 900 and 1000 °C seems to be the same. Therefore, for clarity, only the willemite synthesized at 900 °C is given in Figure 14a.

It was inferred that the intensity of the spectrum is higher in lower calcination temperature between 500 and 600 °C, which tends to swing to lower wavelength upon an increase in the calcination temperature in the violet–blue region. This is because the samples possess lower crystallinity at lower calcination temperature [70]. As the calcination process is increased between 700 and 1000 °C, subsequently, a reduction of the absorption intensity occurs alongside shifting of the peak to the lower wavelength due to an improved crystallinity as evidenced by the XRD reported in this work, and it is in agreement with Babu et al. (2014). The absorption of the samples calcined at 900 and 1000 °C appears to be the same, thus only the one at 1000 °C could be readable in Figure 14. Based on the structural studies of these samples from the XRD, and micrographs, at higher calcination process beyond 700 °C, the reduction in the peak intensity is ascribed to the deteriorating crystal quality of ZnO in the ZnO–SiO_2_ system, and the subsequent formation of willemite crystals [71].

Considering the influence of the dopants, the UV light absorption spectra of the doped samples with various Ce^3+^ amounts calcined at 900 °C are demonstrated in Figure 14b. It is fascinating to note that the intensity of the doped willemite is greater when compared to the undoped sample. Likewise, it was inferred that the absorption edges of the undoped willemite had shifted to longer wavelengths upon the introduction of the Ce^3+^ due to the willemite lattice. These fluctuations can be ascribed to the existence of Ce^3+^ ions in the willemite network, where it might have caused nonbridging oxygen. According to Omar et al. [28], it was suggested that the presence of rare earth ions are responsible for the red shift of the willemite absorption spectra.

### 4.8. Optical Band Gap Analysis

The influence of the calcination temperature and the dopant effects on the optical band gap energy (E_opt_) of the willemite NPs formed between 500 and 1000 °C was determined based on the absorption properties. The E_opt_ values of the willemite nanoparticles formed at different temperatures were evaluated using the Mott and Davis model stated in Equation (2) [11]. The plot is as depicted in Figure 15a. The E_opt_ achieved from the calcination process between 500 and 1000 °C ranged between 3.35 and 5.41 eV. It was observed that when the samples were calcined at temperatures between 500 and 700 °C, E_opt_ value of 3.35–3.27 eV was observed. Conversely, when the samples were further calcined at 800 and 900°C, respectively, the E_opt_ value dramatically increases to 5.25 and 5.32 eV. Then, the E_opt_ value shows a further increase trend of 5.41 eV as the calcination went up to 1000 °C. The impact of the calcination temperature in the sample is such that, at lower heat treatment (500–700 °C), the samples induced a red shift of the electronic absorption edge, which corresponds to E_opt_ of ZnO phase [48]. The dramatic increase in the E_opt_ at higher calcination temperatures is due to the crystallization process as evidenced by the XRD spectra in this work. Other literature similarly reported that the wide E_opt_ obtained at higher calcination in ZnO–SiO_2_ is owed to the collapse in ZnO wurtzite structure and formation of willemite phase [48].

The samples calcined at 900 °C for the doped willemite (1–5 mol% of Ce^3+^ dopant) were used to study the different Ce^3+^ concentration on the host lattice. From the graph shown in Figure 14b, it is fascinating to note that the undoped willemite sample calcined at 900 °C has an energy band gap of 5.32 eV. The energy decreases from 5.21 to 5.00 eV with the introduction of 1–5 mol% of Ce^3+^ dopant into the willemite crystal system. This effect is explained by a phenomenon of Burstein–Moss shift which is triggered by heavy hole concentration and nonbridging oxygen as the density of the impurities atom increases [72]. Additionally, the band gap energy may decrease as well due to high doping densities, which is caused by overlapping of an electron wave function with the impurity atoms forcing the energies to compose an energy band rather than a discrete level [73]. The obtained E_opt_ values for undoped and the doped samples are presented in Table 4 and Table 5, respectively.

Consequently, it can be summarized that the optical band gap was owed to the direct allowed transition. Thus, the E_opt_ values achieved in the current work conform with the research conducted by other researchers [24,48,74,75,76].

### 4.9. Optical Emission Spectra

The calcination temperature and Ce^3+^ ions dopant effects on the optical emission properties for the samples calcined between 500 and 1000 °C was studied using PL spectroscopy excited at 350 nm excitation wavelength. The spectra in Figure 16 showed the emission properties of willemite NPs at various calcination temperatures. The behavior of the PL spectrum of a material is generally peculiar and depends on the synthesis method employed [77,78,79,80]. Another important factor affecting the PL emission is the increase in the calcination process. According to Babu et al. [48], the calcination temperatures had effects on the intensity of the PL spectra. This is by the XRD spectra discussed earlier; the higher calcination process would increase the crystallinity and particle size of materials. These are considered the contributory factors to the reduction in nonradiative recombination routes for the electrons and holes, hence, influencing the increase in the emission intensity of the materials [81,82,83].

From the spectra in Figure 16a, several emission peaks were observed at violet, blue, and green regions as follows; at 423, 447, 484, and 530 nm wavelengths, respectively. In this light, the defect emission peaks at 423 and 447 nm, respectively, are accredited to oxygen defects in the violet–blue region, and it is referred to as the blue emission [84,85,86]. The strong emission at 484 nm is attributed to the zinc interstitials [86]. The emission peak observed in the green region roughly at 530 is owed to the apparent transition of the electrons amid the valence and conduction [87]. The increase in the emission peak as the calcination temperature increases is due to the enhancement in the crystal quality of the materials and the strain induced in the interface of ZnO and SiO_2_ [86,87]. The peak in the blue region at 480 nm observes a little shift to 485 nm between 800 and 1000 °C. This behavior was ascribed to the enhancement in crystal feature of the material and subsequent deterioration of the ZnO crystal structure to form willemite [86].The impact of the Ce^3+^ ions dopant (1–5 mol%) on the optical emission of willemite phosphor is taken into consideration, from the emission spectra shown in Figure 16b, the doped samples demonstrated a strong emission peak at 400 nm coming from the guest ion (the dopant) which is due to the transition of an electron from localized Ce_2f_ state to the valence band of O_2p_ [55]. It is evident that the addition of cerium dopant in the willemite matrix decreased the intensity of the green emission at 530 nm and the blue emission at 485 nm. This can be owed to the fact that the energy levels of Ce^3+^ affect the crystal lattice [39]. Additionally, an emission peak at 422 nm ascribed to the O_2_ defects had disappeared due to the introduction of the dopant. Moreover, the emission intensity progressively increased with the addition of the dopant concentration [38,55].

## 5. Conclusions

In summary, this study reported the synthesis of undoped and Ce^3+^ doped willemite phosphors calcined at different temperatures based on polymer thermal treatment method. The XRD revealed the amorphous structure of the uncalcined sample and showed a crystal phase of willemite NPs after calcination between 800 and 1000 °C. The resulting XRD and EDX analyses outcomes clearly showed that the Ce^3+^ ion was doped into the willemite crystal lattice. The Raman and FT–IR spectra reassured the existence of organic source in the samples before the calcination and crystallization of samples upon the heat treatment with the stronger SiO_4_ and ZnO_4_ absorption bands. The mean crystallite size and particles size of the samples increases from 36.7 to 23.8 nm with the addition of Ce^3+^ concentration. The UV optical spectra for willemite reveals an absorption in the violet–blue region (blue-shift) and their calculated bandgap energies decreases from 5.21 to 5.00 eV with the introduction of 1–5 mol% of Ce^3+^ dopant into the willemite crystal system. The doped samples exhibited a strong PL emission peak at 400 nm originating from the dopant which is attributed to the transition of an electron from localized Ce_2f_ state to the valence band of O_2p_. The current findings may pave the way to further understand the effects of calcination and Ce^3+^ ions concentration on the optical and structural properties of willemite semiconductor NPs and wide band gap energy of the material. It may also have numerous possible applications for imminent LED and other optoelectronic lighting devices.

## Figures and Tables

**Figure 1 materials-14-01118-f001:**
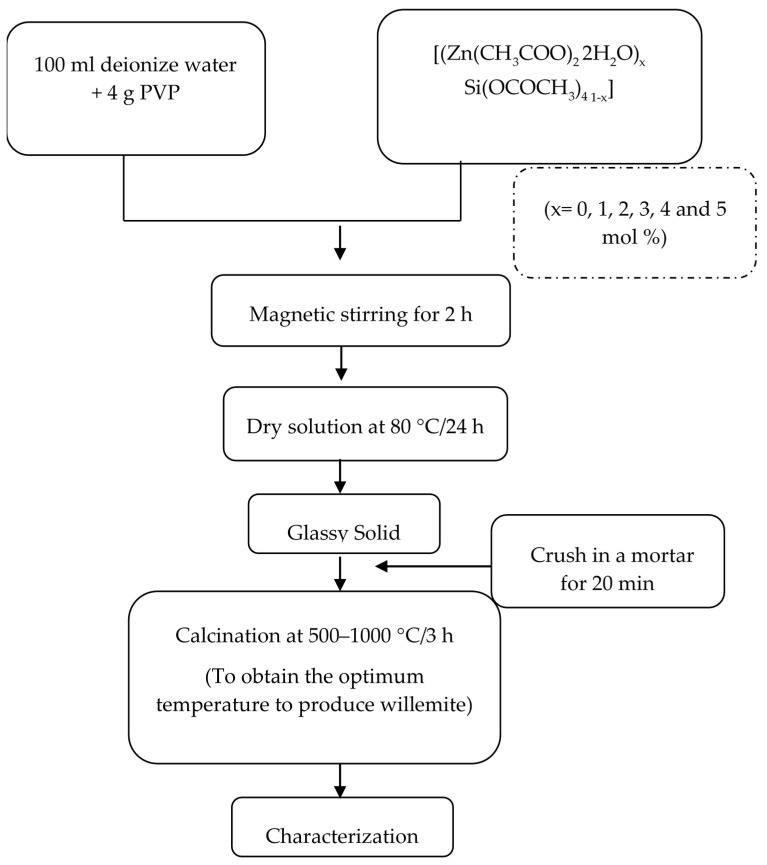
Schematic diagram of the synthesis process.

**Figure 2 materials-14-01118-f002:**
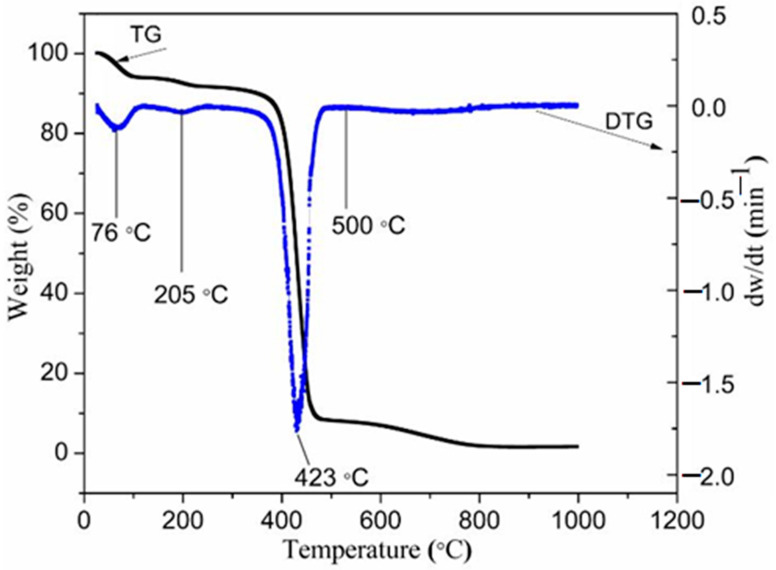
Thermogravimetric (TG) and its derivative (DTG) curves for PVP at a heating rate of 10 °C/min.

**Figure 3 materials-14-01118-f003:**
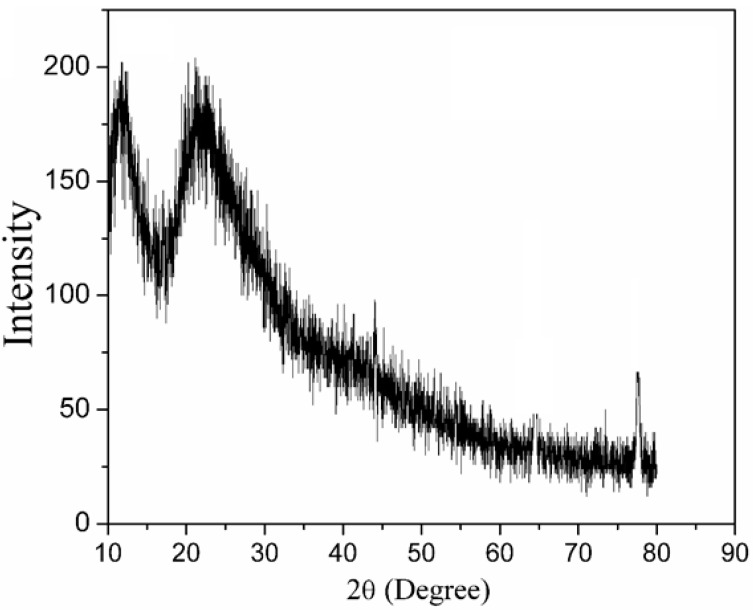
X-ray diffraction pattern of the uncalcined sample at room temperature.

**Figure 4 materials-14-01118-f004:**
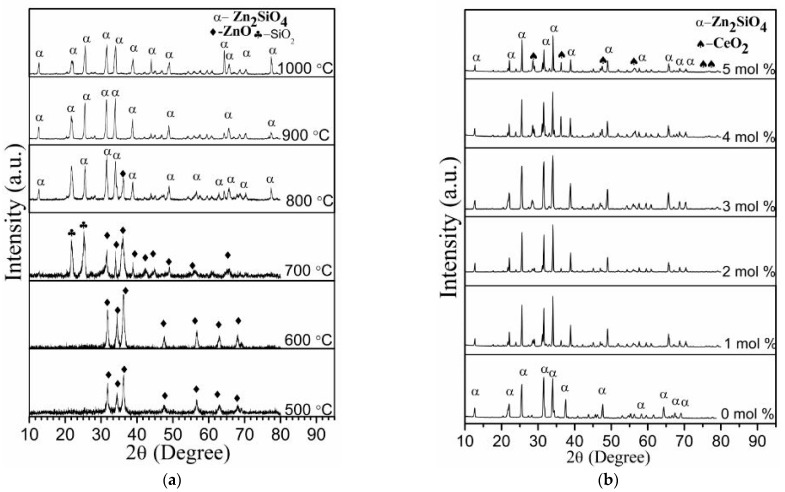
(**a**) XRD spectral patterns of willemite nanoparticles (NPs) at various calcination temperatures between the range 500–1000 °C, (**b**) XRD patterns of willemite with different Ce^3+^ doping concentration calcined at 900 °C.

**Figure 5 materials-14-01118-f005:**
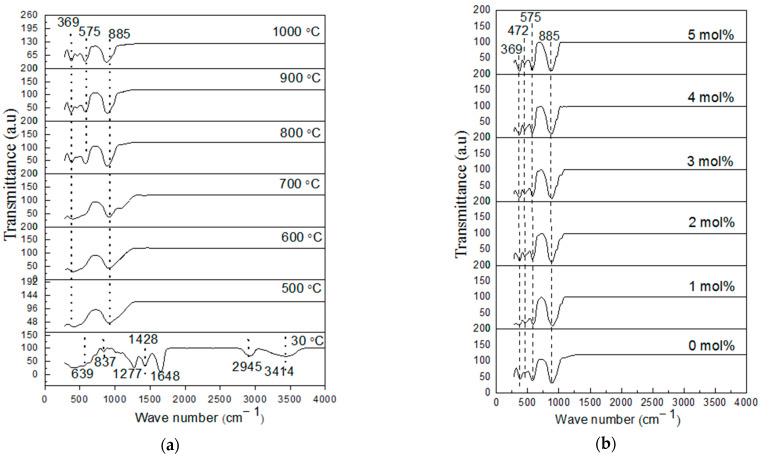
(**a**) FT–IR spectra of willemite NPs calcined different temperature; (**b**) FT–IR patterns of willemite with different Ce^3+^doping concentrations calcined at 900 °C.

**Figure 6 materials-14-01118-f006:**
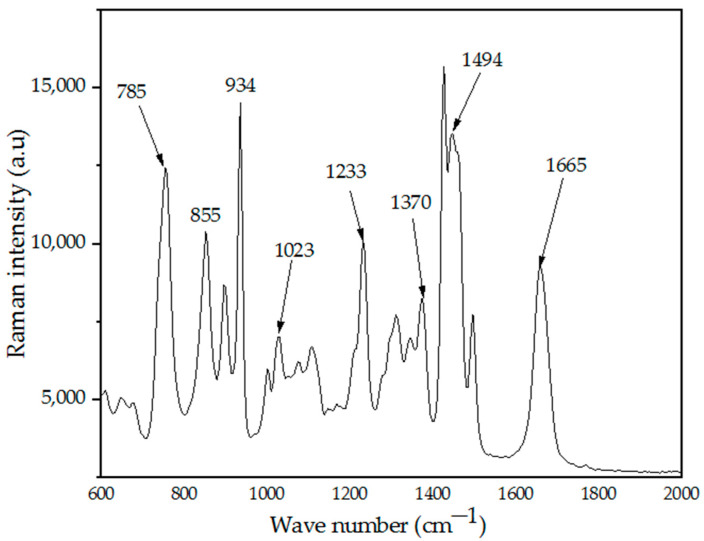
Raman spectrum of the sample before the calcination process.

**Figure 7 materials-14-01118-f007:**
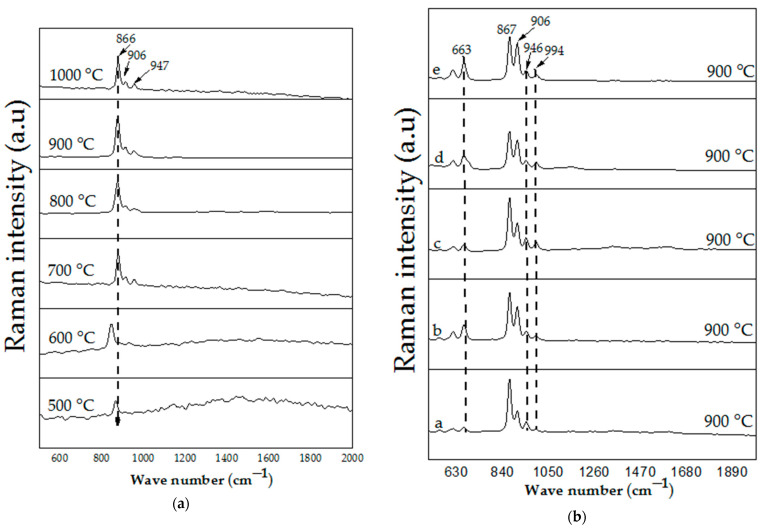
(**a**) Raman spectra of willemite NPs calcined at different temperatures; (**b**) FT–IR patterns of willemite with different Ce^3+^doping concentrations calcined at 900 °C; (a) 1 mol%, (b) 2 mol%, (c) 3 mol%, (d) 4 mol%, and (e) 5 mol%.

**Figure 8 materials-14-01118-f008:**
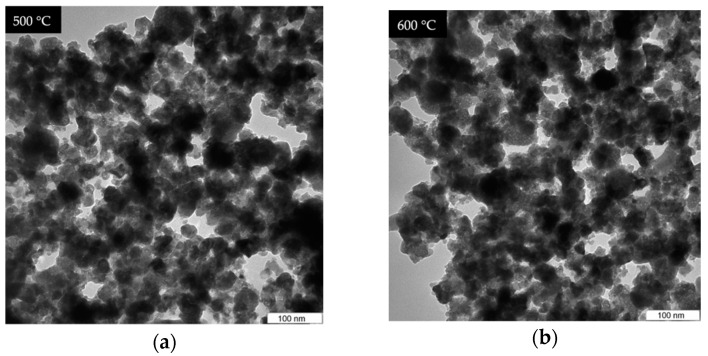
TEM analysis of willemite NPs calcined at different temperatures: (**a**) 500, (**b**) 600, (**c**) 700, (**d**) 800, (**e**) 900, and (**f**) 1000 °C.

**Figure 9 materials-14-01118-f009:**
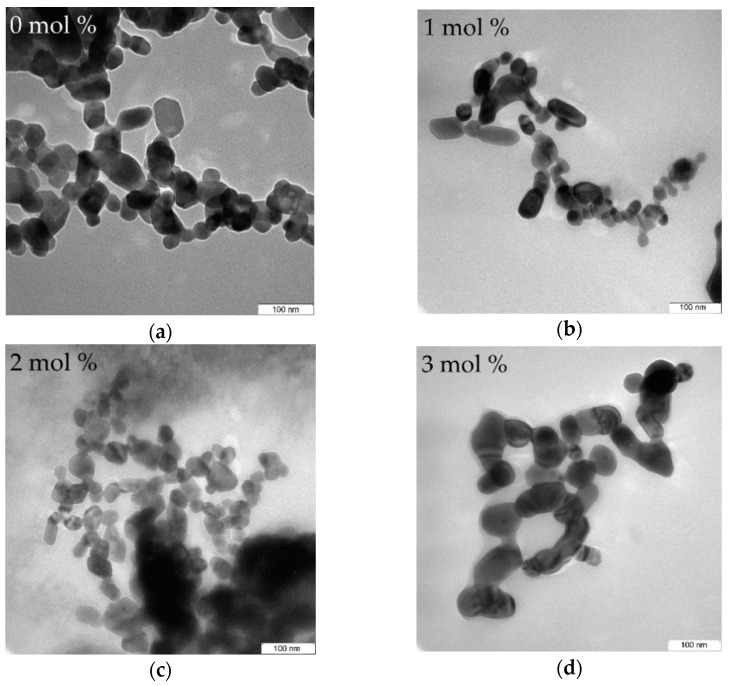
TEM images of willemite NPs size distribution calcined at 900 °C with different Ce^3+^ doping concentrations; (**a**) 0 mol%, (**b**) 1 mol%, (**c**) 2 mol%, (**d**) 3 mol%, (**e**) 4 mol%, and (**f**) 5 mol%.

**Figure 10 materials-14-01118-f010:**
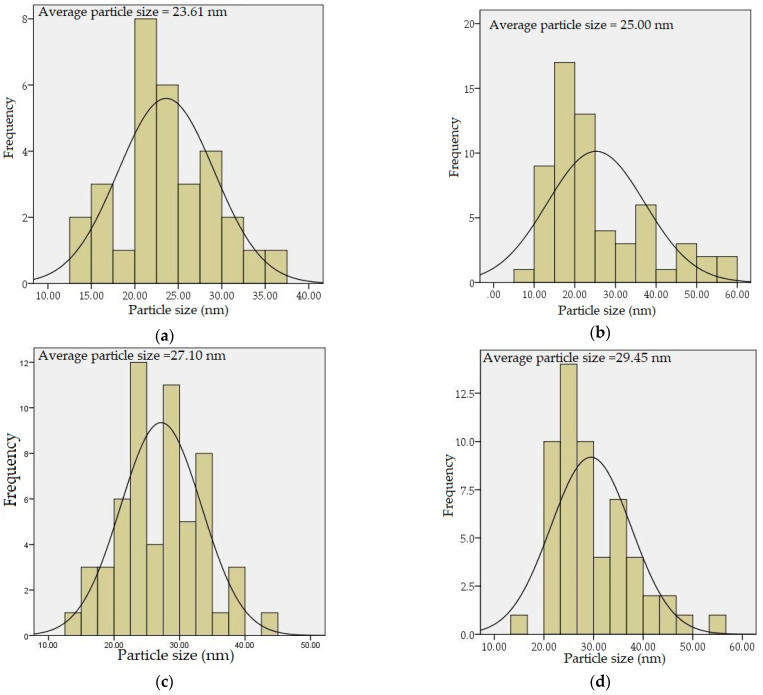
Willemite NPs size distribution calcined at 900 °C with different Ce^3+^ doping concentrations; (**a**) 0 mol%, (**b**) 1 mol%, (**c**) 2 mol%, (**d**) 3 mol%, (**e**) 4 mol%, and (**f**) 5 mol%.

**Figure 11 materials-14-01118-f011:**
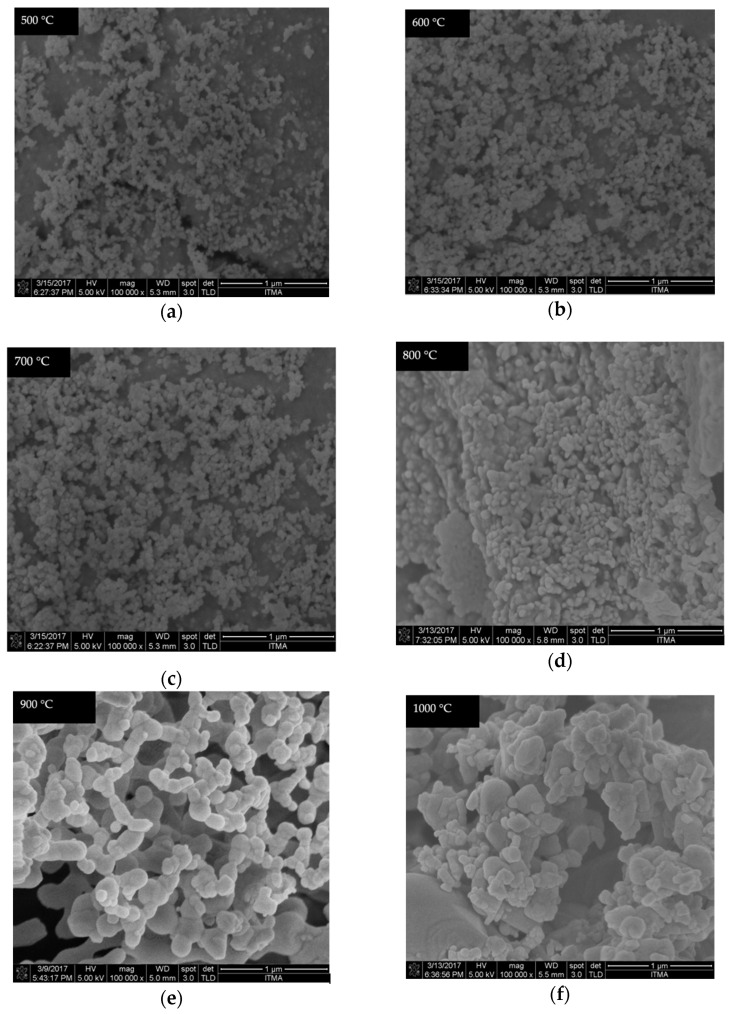
FESEM micrographs of the undoped willemite NPs calcined at different temperatures: (**a**) 500, (**b**) 600, (**c**) 700, (**d**) 800, (**e**) 900, and (**f**) 1000 °C.

**Figure 12 materials-14-01118-f012:**
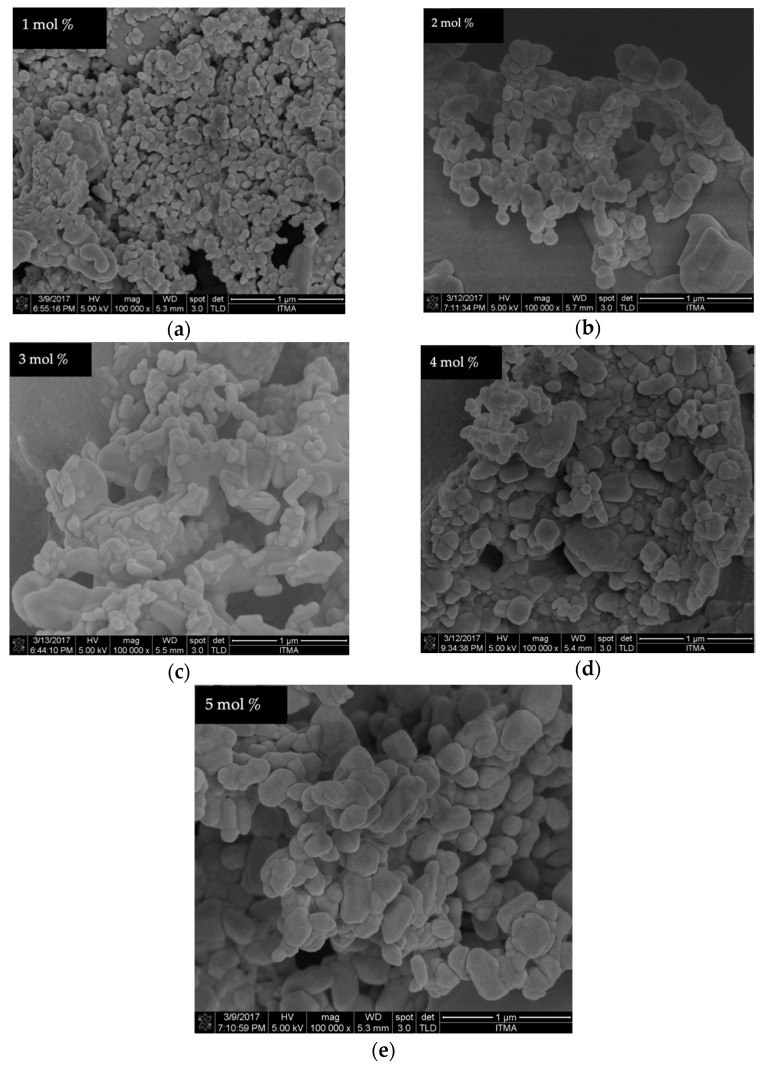
FESEM images of willemite NPs calcined at 900 °C with different Ce^3+^ doping concentrations; (**a**) 1 mol%, (**b**) 2 mol%, (**c**) 3 mol%, (**d**) 4 mol%, and (**e**) 5 mol%.

**Figure 13 materials-14-01118-f013:**
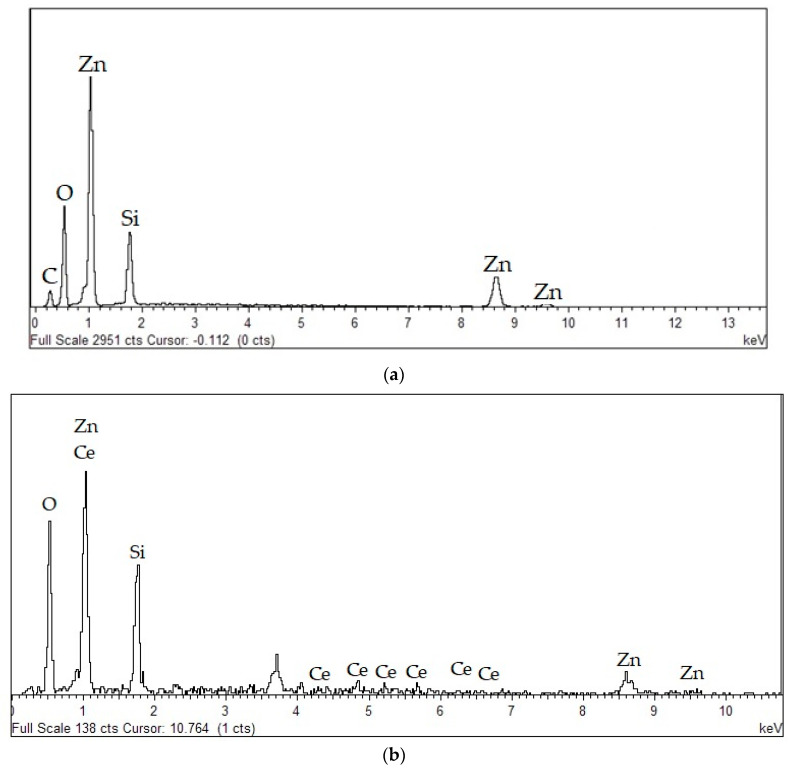
EDX spectra of (**a**) undoped willemite; (**b**) willemite: Ce^3+^ 5 mol% calcined at 900 °C.

**Figure 14 materials-14-01118-f014:**
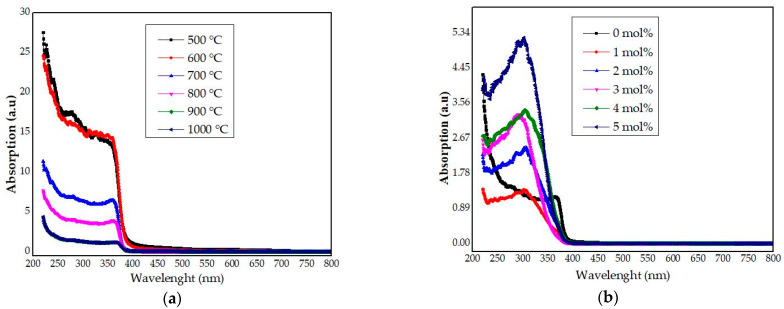
UV absorption spectra of (**a**) willemite NPs calcined at different temperatures, and (**b**) willemite doped with different Ce^3+^ concentrations calcined at 900 °C.

**Figure 15 materials-14-01118-f015:**
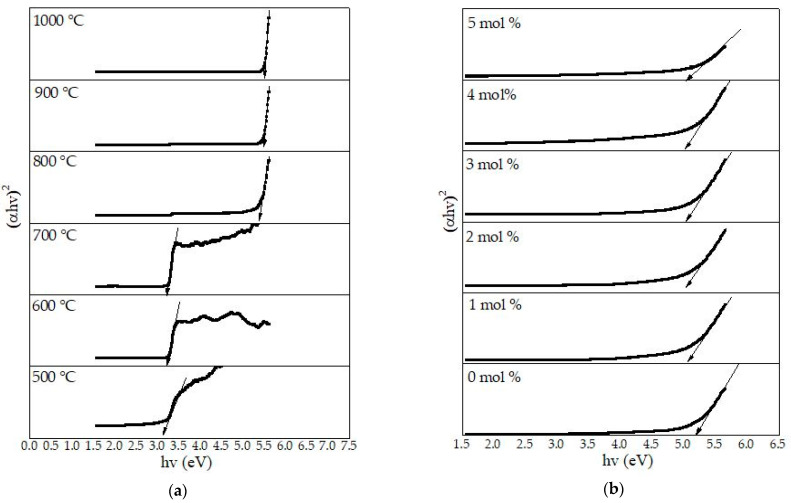
Plot of ${\left(\propto hv\right)}^{2}$ versus hv for (**a**) willemite NPs calcined at different temperatures, and (**b**) willemite doped with different Ce^3+^ concentrations calcined at 900 °C.

**Figure 16 materials-14-01118-f016:**
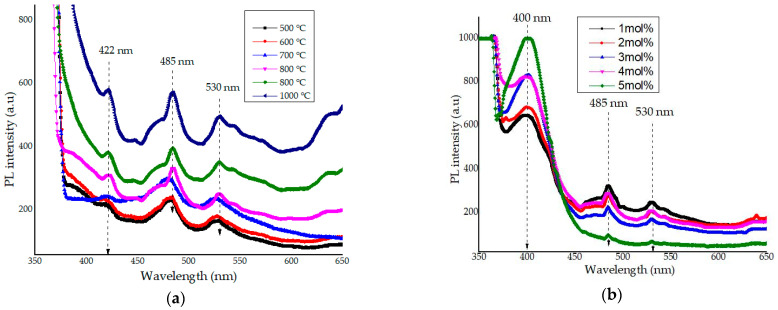
PL emission spectra of (**a**) willemite NPs calcined at different temperatures, and (**b**) willemite doped with different Ce^3+^ concentrations calcined at 900 °C.

**Table 1 materials-14-01118-t001:** Summary of the structural and optical properties of willemite doped with Ce^3+^ calcined at 900 °C with different doping concentration.

Dopant Concentration (mol%)	D_XRD_ (nm)	D_TEM_ (nm)	Optical Band Gap (eV)
0	22.55	23.61	5.32
1	24.40	25.00	5.21
2	26.30	27.10	5.08
3	28.29	29.45	5.05
4	30.40	31.50	5.02
5	33.35	34.02	5.00

**Table 2 materials-14-01118-t002:** FT–IR absorption bands and their corresponding assignments.

Wave Number (cm^−1^)	Assignment of the Vibration Mode
379–441	Zn–O asymmetric stretching vibration
580	ZnO_4_ symmetric stretching vibration
885–889	SiO_4_ symmetric stretching vibration
900–989	SiO_4_ asymmetric stretching vibration
639–641	C–N=O bending vibration
837	C–C ring vibration
1277	C–N stretching vibration
1402–1460	C–H bending vibration of the methylene group
1648	C=O stretching vibration
2945	C–H vibration
3414	N–H bending vibration

**Table 3 materials-14-01118-t003:** Assignment and interpretation of the Raman absorption bands.

Wave Number (cm^–1^)	Vibrational Mode Assignment
300–700	Bending vibration of SiO_4_ group
800–1100	Stretching vibration of SiO_4_ group
868	Crystalline Zn_2_SiO_4_ vibration peak
758	Ring C–C vibration
885	C–C stretching vibration
934	C–C Ring breathing
1023	C–C Back bone
1370	C–H Deformation
1494	CH_2_ Scissors
1675	C=O

**Table 4 materials-14-01118-t004:** Willemite NPs optical band gap at various calcination temperatures.

Temperature (°C)	500	600	700	800	900	1000
E_opt_ (eV)	3.35	3.30	3.27	5.25	5.32	5.41

**Table 5 materials-14-01118-t005:** Optical band gap for willemite doped with different Ce^3+^ concentrations calcined at 900 °C.

Dopants mol%	0	1	2	3	4	5
E_opt_ (eV)	5.32	5.21	5.08	5.03	5.02	5.00

## Data Availability

According to the Universiti Putra Malaysia (Research) Rules 2012; All data regarding this work could be obtained from the office of Deputy Vice-Chancellor (Research and Innovation).

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
