# Peer review of "Polymer Thermal Treatment Production of Cerium Doped Willemite Nanoparticles: An Analysis of Structure, Energy Band Gap and Luminescence Properties"

_materials, 2021, doi:10.3390/ma14051118_

Round 1
Reviewer 1 Report
In this manuscript, the authors synthesized the Ce3+ doped willemite nanoparticles via polymer thermal treatment method and made the analysis on the properties of the prepared willemite nanoparticle in detail. However, the novelty is not high. I recommend publication to the journal after the authors made appropriate revisions in the manuscript.
- The synthetic route of nanoparticle is very important in this article. In order to display more intuitive understanding of the material preparation for reader, a schematic diagram is necessary.
- The authors declaimed “The nanoparticle size increase with respect to increase in the cerium ion by necking between the neighboring particles. This is attributed to substitution of the Ce3+ ions for Zn ions which distorts the willemite structure”. What’s the fundamental reason for the increased nanoparticle size? I don’t think the authors explain it clearly. Please give more discussion.
- The calculated Eopt is wrong, please check.
- The authors said “the material may have key potential applications for future LED and other optoelectronic lighting devices”. I can't find the promising application for optoelectronic light devices. The author need discussion more on the differences between inorganic Ce3+ doped willemite nanoparticle and commercial conjugated organic materials in optoelectronic field.
Author Response
Dear Sir
The authors will like to use this opportunity to express their appreciation for the useful recommendations, suggestions, and corrections to the manuscript.
Kindly find attached the author's response.

Reviewer 2 Report
In this paper, Alibe and coworkers propose an innovative approach for the design of Ce3+ doped willemite nanoparticles followed by a thorough study of their properties.
Samples were synthetized using polymer thermal treatment method through calcination of PVP based mixture of precursors. The effects of the calcination temperatures and the dopant amounts are studied through TGA, XRD, TEM and FESEM. The study is then followed by optical analysis with UV/Vis, Band Gap and emission measurements.
The paper is well written. Although the novelty in the field remains weak, the study is thorough enough to bring interesting results to the readers. Therefore the work can be published with few corrections.
ATG/DSC section
The curve is not flat after 500°C and a significant weight loss (7-8%) can still be observed between 450 and 800°C. The authors should mention its origin and explain the phenomenon.
XRD section
For XRD the graph can not be described as a spectrum but a diffractogram (ex line 176 p5).
In fig(a) XRD patterns exhibit a signal around 36° attributed to Cerium oxide in the 5% doped sample. This signal tends to be strong for 4%molar, small in the case of 1 and 2% mol and completely disappear for 3 %mol. Authors should explain this behavior.
TEM section
For Fig 7, b, d and e : the temperatures are not readable.
UV/Vis section
The authors should mention if the particles were analyzed in suspension or as a powder.
In Fig 13 a, the spectrum corresponding to the sample heated at 900°C does not appear.
Author Response
Dear Sir,
The authors will like to use this opportunity to express their appreciation for the useful recommendations, suggestions, and corrections to the manuscript.
Find attached the author's response.
Thank you.
Best Regards

Round 2
Reviewer 1 Report
In this revised manuscript, the authors have well addressed my concerns. I would like to recommend publication of this manuscript.